# Association of MMP-2 and MMP-9 Polymorphisms with Diabetes and Pathogenesis of Diabetic Complications

**DOI:** 10.3390/ijms231810571

**Published:** 2022-09-12

**Authors:** Beata Gajewska, Mariola Śliwińska-Mossoń

**Affiliations:** 1Students Scientific Society for Specialist Biological Analyzes, Faculty of Pharmacy, Wroclaw Medical University, 50-556 Wroclaw, Poland; 2Department of Medical Laboratory Diagnostics, Division of Clinical Chemistry and Laboratory Haematology, Faculty of Pharmacy, Wroclaw Medical University, 50-556 Wroclaw, Poland

**Keywords:** metalloproteinase 2, metalloproteinase 9, diabetes

## Abstract

Type 2 diabetes mellitus (T2D) affects millions of people around the world, and its complications have serious health consequences. In addition to external factors, the causes of morbidity and increased risk were also sought in the variability of the human genome. A phenomenon that can answer these questions is the occurrence of single-nucleotide polymorphisms (SNP). They constitute a field for research into genetic determinants responsible for the increase in the risk of the discussed metabolic disease. This article presents the outline of two enzymes: metalloproteinases 2 and 9 (MMP-2, MMP-9), their biological activity and the effect caused by differences in individual alleles in the population, as well as the reports on the importance of these DNA sequence variations in the occurrence of diabetes mellitus type 2 and associated conditions. The results of the conducted research indicate a relationship between two MMP-2 polymorphisms (rs243865, rs243866) and two MMP-9 polymorphisms (rs3918242, rs17576) and the presence of T2D. This could offer a promising possibility to use them as predictive and diagnostic markers. However, due to the low number of reports, more research is needed to clearly confirm the link between these SNPs and diabetes.

## 1. Properties, Functions and Regulation of the MMP

### 1.1. Matrix Metalloproteinases—General Characteristics

Matrix metalloproteinases (MMPs) belong to the family of zinc-dependent endoproteases. Their functions are based on the remodeling and degradation of protein components belonging to the extracellular matrix (ECM) [1]. The actions of these enzymes in a number of processes that cells undergo—proliferation, migration, differentiation—are strictly controlled by mechanisms of transcription, translation and by inhibitors. Disrupting this balance can lead to negative health effects consisting in unregulated work of metalloproteinases [2]. In addition to physiological processes, which include participation in angiogenesis, embryogenesis, wound healing and platelet aggregation, MMP is involved in a number of pathological changes such as the development of cardiovascular diseases, disorders of the musculoskeletal system, development of cancer and inflammatory processes [3].

### 1.2. Gelatinase

From the family of MMP enzymes, in functional terms, we can distinguish gelatinases as those having a catalytic domain adapted in structure to gelatin hydrolysis [4]. Representatives of this group are two metalloproteinases: MMP-2 (72 kDa gelatinase, Gelatinase-A) and MMP-9 (Gelatinase-B, 92 kDa gelatinase).

In addition to physiological processes, research has shown their important role in the course of cancer invasion and metastasis as well as other degenerative diseases, and the polymorphisms of genes encoding these MMPs are the subject of research on possible differences in susceptibility to certain diseases and their diversified course and development [5].

### 1.3. MMP-2 and MMP-9

In the structure of enzymes (Figure 1) we can distinguish a catalytic site: three type II fibronectin repeats, which enable enzymes to bind denatured collagen and elastin [4,6]. This property is involved in a number of physiological and pathological processes, largely in the progression and metastasis of various types of cancer.

The mechanisms mediated by gelatinases, which may contribute to the development of neoplasms, are the degradation of elements of the extracellular matrix (Figure 2). One of these components—collagen IV—is part of the basal membrane, the degradation of which has a direct impact on the progression of metastases [7,8,9,10]. Another factor contributing to the formation of tumors is angiogenesis, which originates from the influence of these enzymes on the migration of endothelial cells, which is a key feature of angiogenesis in neoplasia [7,11,12]. It is caused by the increase of the vascular endothelial growth factor and the release of the bioactive factor VEGF, a potent stimulator of angiogenesis (Figure 2) [13,14,15].

MMP-2’s gene locus occupies the position on the chromosome 16q13-q21. RNA expression occurs in all tissues, with particular emphasis on the gallbladder, bladder, smooth muscle tissue and the endometrium [16,17,18,19]. Protein expression shows a pattern in cytoplasmic expression in trophoblastic and decidual cells of placenta, macrophages and alveolar cells of the respiratory epithelium, urothelial cells of the urinary bladder, ciliated cells of the Florian tube, stromal cells of the endometrium as well as the endothelial and stromal cells in most tissues [19]. The disorder in which the coding gene is directly involved is multicentric osteolysis-nodulosis-arthropathy (MONA) disorders [20,21]. The functional polymorphism of this gene is also suspected to influence the risk of the metabolic syndrome. The researchers pointed to the variants −1575 G/A (rs243866), −168 G/T and also −1306 C/T (rs243865) as those that may be significant in determining the factors of this disorder, the complication of which may be the development of type 2 diabetes and cardiovascular disease [22,23,24,25].

MMP-9′s gene locus occupies the position on the chromosome 20q11.2-q13.1. RNA expression has been detected in many types of tissues with particular specificity for bone marrow and lymphoid tissue [16,17,18,19]. Protein expression occurs as selective nuclear and cytoplasmic expression in proximal tubules of kidney, cells in the red and white pulp of the spleen, hematopoietic cells of bone marrow, preleptotene spermatocytes and the spermatogonia cells of the testis and non-germinal center cells of the appendix, lymph nodes and tonsils [19]. The literature contains examples of disorders involving the mutation of the MMP gene, including metaphyseal anadysplasia in which it occurs together with the MMP-13 gene [26,27]. As in the case of MMP-2, here the functional polymorphism of the gene encoding this metalloproteinase has also been associated with the occurrence of metabolic syndrome. The presence of SNP −1562 C/T (rs3918242) was associated with an increased risk of this disorder [28].

## 2. MMPs Polymorphisms Associated with Type 2 Diabetes

Research on the polymorphisms of metalloproteinase genes indicate their participation in many diseases. In the direct case of type 2 diabetes, the previous reports mention two enzymes from this family: MMP-2 and MMP-9.

### 2.1. MMP-2

The studies reported two single-nucleotide polymorphisms of the promoter of the MMP-2 gene, −1306 C/T (rs243865) and −1575 G/A (rs243866), as those that may be associated with the risk of developing type 2 diabetes [29]. In addition to the publications related to T2D, these SNPs have also been linked to other disorders, −1306 C/T with esophageal cancer in the Asian population and −1575 G/A in single reports of risk of macular degeneration and endometriosis [30,31,32].

The −1306 C/T polymorphism (rs243865) is located in the CCACC box, where it leads to a decrease in promoter activity by disrupting the Sp1 binding side [33,34,35]. Hence, the presence of the homozygote genotype with the T allele will lead to a lower level of MMP-2 and, consequently, a protective effect against the negative aspect of gelatinase A. On this basis, it is assumed that people with the CC genotype may show an increased susceptibility to the incidence of type 2 diabetes by overexpression of the MMP-2 protein [29].

In the case of the −1306 C/T polymorphism (rs243865), it is suggested that the functional effect of the polymorphism differs depending on the cell type and is related to the presence of estrogen receptors. For example, in the MCF-7 cell study, the G allele enhanced transcription, while the A allele decreased transcription [33,36]. Another study found that the polymorphism was not functional in cells without estrogen receptors. Another explanation that was presented was the regulation of transcript through binding sites for transcription factors, where the A allele is suggested to be in linkage disequilibrium with other regulatory regions in the MMP-2 gene’s promoter. Consequently, this leads to a change in the level of the MMP-2 protein compared to the G variant [33,35,36,37].

Regarding the frequency of occurrence of a given SNP, the current data for the general population for −1306 C/T (rs243865) are 0.20139, with the highest frequency being the European population −0.24479 and the smallest in the African population with African ancestry, 0.025. For −1575 G/A (rs243866), the data for the general population is 0.229833, with the highest frequency also in the European population at −0.242161 and the lowest in the African ancestry population at −0.022 (Table 1) [38].

Research on the connection of the MMP-2 polymorphism and the incidence of type 2 diabetes is one publication in which the authors focus on the −1306 C/T (rs243865) and −1575 G/A (rs243866) polymorphisms (rs243866) as well as rs2285053 (−735 C/T) and rs9923304. There was no correlation between rs2285053 (−735 C/T) and rs9923304 and T2D. On the other hand, −1306 C/T (rs243865) and −1575 G/A (rs243866) showed the potential to reduce the risk of developing T2D (Table 2).

### 2.2. MMP-9

The studies reported two single-nucleotide polymorphisms of the promoter of the MMP-9 gene, −1562 C/T (rs3918242) and +279 A/G (rs17576), as those that may be associated with the risk of developing type 2 diabetes [39,40]. In addition to the publications related to T2D, these SNPs have also been linked to other disorders, such as −1562 C/T with colorectal, lung and breast cancer, endometriosis, ischemic stroke, asthma, multiple sclerosis and chronic obstructive pulmonary disease and +279 A/G with glaucoma [41,42,43,44,45,46,47,48].

SNP −1562 C/T (rs3918242), which is a functional polymorphism in the promoter region, has a direct effect on increasing the degree of gene transcription by loss of nuclear protein binding. The presence of the T allele causes an approximately 1.5-fold increase in the activity of the promoter and thus the increase in the level of the MMP-9 protein and its consequences. It is postulated that this may result in genetic susceptibility in patients with type 2 diabetes and altered matrix depositions in the wounds of diabetic patients [39,49,50].

The MMP-9 A/G polymorphism (rs17576) lies in the substrate binding region and is a functional variant that changes the conformation of the MMP-9 protein by replacing the uncharged amino acid glutamine with the positively charged amino acid arginine. This results in a change in MMP-9 substrate binding and enzyme activity, and also reduces the binding affinity of type IV collagen [4,51,52,53,54].

Regarding the frequency of occurrence of a given SNP, the current data for the general population for −1562 C/T (rs3918242) are 0.17402, with the highest frequency being in the South Asian population at −0.327, and the smallest frequency being in Asian individuals, excluding South or East Asian, at −0.04. For +279 A/G (rs17576), the data for the general population is 0.355297, with the highest frequency also in the Asian population at −0.755 and the lowest in the Latin American individuals with mostly European and Native American ancestry at −0.2256 (Table 1) [38].

Studies on the connection of MMP-9 polymorphism and type 2 diabetes are two publications in which the authors focus on polymorphisms −1562 C/T (rs3918242) and +279 A/G (rs17576). The publication on −1562 C/T (rs3918242) showed that a given polymorphism may be significant with respect to the susceptibility to type 2 diabetes. Another publication was based on +279 A/G (rs17576) and COL4A3 (G/T) and TIMP-1 (A/G). There was no significant correlation between T2D and TIMP-1 (A/G), while +279 A/G (rs17576) and COL4A3 (G/T) were positively correlated with type 2 diabetes, with COL4A3 (G/T) as a protective factor and +279 A/G (rs17576) as risk factor (Table 2) [39,40].

In the case of the −1562 C/T (rs3918242) polymorphism, 730 people participated in the case-control study, including 353 patients diagnosed with type 2 diabetes (and 267 healthy people in the control group). Participants in the study belonged to the same ethnic group and were recruited from the general north Indian population residing in and around the Varanasi district, Uttar Pradesh, India [39].

The case-control study on the +279 A/G polymorphism (rs17576) involved 240 patients, including 120 confirmed type 2 diabetics and 120 healthy individuals, both consisting of Iranian patients [40]. 

The characteristics of all groups are summarized in Table 3 [29,39,40]. From a comparison of the basic factors characterizing the control subjects, it can be seen that they had a lower BMI that was normal or slightly above normal, while the diabetic group was overweight [29]. In the studies for which the average age of the subjects was given, its values were similar between the control and T2D groups [29,40], while the difference between the individual studies was about 3.89 years in the control group [29,40] and 9.19 years [29,39], 4.03 years [29,40] and 5.16 years [39,40] in the groups of patients with diabetes. If we take into account the time from the onset of the disease, we notice that the age of the patients at the time of diagnosis was similar (47 years [29], 45.27 years [39], 46.66 [40], respectively).

In the study concerning MMP-2, an additional criterion for the selection of the subjects was used; other types of diabetes, autoimmune diseases and a positive anti-GAD, anti-IA2 or ICA autoantibody response were assumed as excluding factors [29].

## 3. Gelatinase Polymorphisms in Diseases Accompanying Type 2 Diabetes

In addition to the direct impact on the incidence of type 2 diabetes, studies have also looked for links to diseases that are components of the metabolic syndrome and complications of T2D.

They have possible effect on diabetic retinopathy [34,55,56,57], diabetic nephropathy [58,59,60,61], diabetic foot ulcers [39] and macroangiopathy [62] as well as cardiovascular comorbidity [63] (Table 4).

Assessment of the effect of SNPs may determine their usefulness as a marker of disease development. Genetic determinants are an interesting direction that can be explored for a more complete picture of patient health conditions and extended diagnostics.

## 4. Conclusions

The research results presented by the authors show a relationship between individual SNPs and the occurrence of type 2 diabetes. Important factors in the development of vascular complications in T2D are the increased glycation, degradation and/or accumulation of elastin and collagen in the vascular wall. MMPs, which hydrolyze the protein components of the vascular extracellular matrix, are actively involved in this process. Deregulation of gelatinase activity is associated with vasculitis, remodeling and fibrosis and may contribute to the pathophysiology of diabetes complications. Therefore, the links between individual SNPs reported in this work also for related TD2 diseases are very important. This provides the basis for the claim that MMP-2 and MMP-9 can be considered as potential markers useful in predicting the risk and course of T2D. This would enable the diagnosis to be deepened and would allow for a more complete picture of the health of a genetically burdened patient predisposed to metabolic syndrome, T2D and complications even before their occurrence. The specificity of polymorphisms and the differences in their frequency between ethnic groups may also be the basis for further research into the relationship between diabetes and the genome. However, the small number of presented results makes it impossible to draw unequivocal conclusions. Further research is needed, with the extension of the study groups to all ethnic groups due to the different frequency of alleles in each of them.

## Figures and Tables

**Figure 1 ijms-23-10571-f001:**
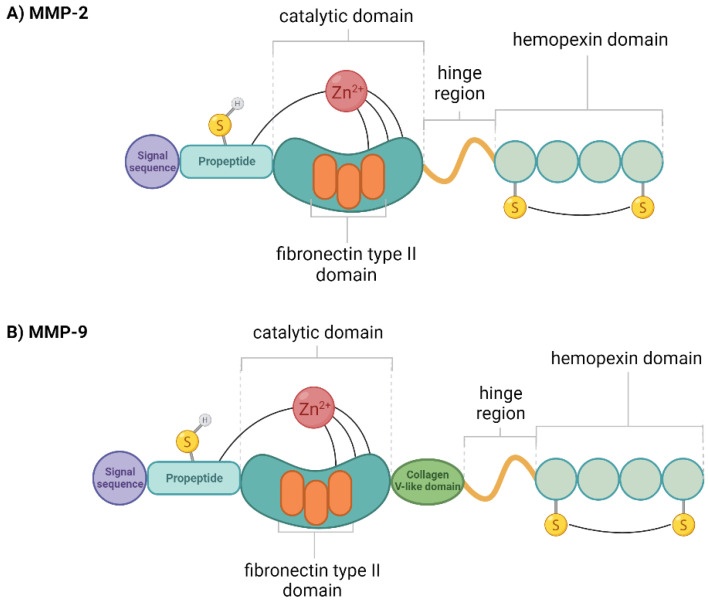
Diagram of the domain structure of (**A**) gelatinase-A (MMP-2) and (**B**) gelatinase-B (MMP-9). Created with BioRender.com.

**Figure 2 ijms-23-10571-f002:**
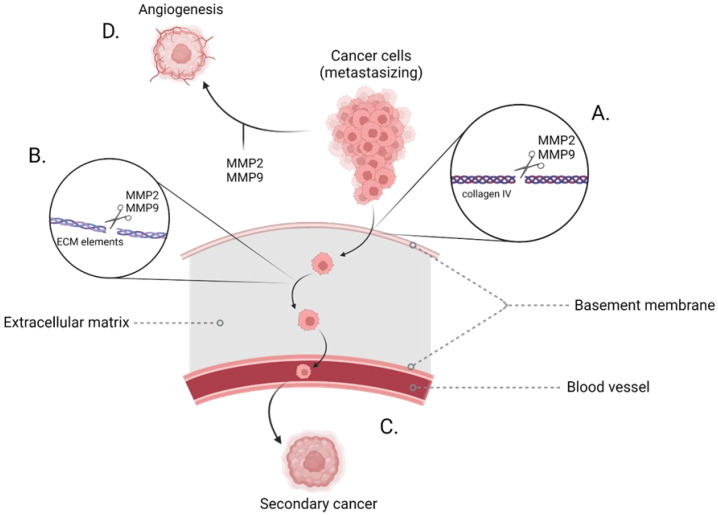
Diagram of the gelatinase action process in the processes of tumor metastasis. (**A**) Hydrolysis of α chains collagen bonds in collagen IV triple-helix building the basement membrane. (**B**) Degradation of the protein components of the extracellular matrix. (**C**) Catalysis of the components of the basement membrane and extracellular matrix causes their degradation and enables the migration of cancer cells through the epithelium to the blood vessels from where they move around the body, finally creating secondary cancers in the sites of metastasis. (**D**) Gelatinases have a positive effect on the tumor angiogenesis processes. Created with BioRender.com.

**Table 1 ijms-23-10571-t001:** The frequency of SNP for of individual polymorphisms of MMP-2 and MMP-9 for general population and the ethnically groups.

Frequency of Alternative Allele among Population (=1−the Frequency of the Reference Allele)													
Gelatinase	SNP	Alternative Allele	Total	All Africans	African American	Individuals with African Ancestry	Asian Individuals Excluding South Asian	East Asian	South Asian	Asian Individiuals Excluding South or East Asian	European	Latin American Individiuals with Afro-Caribbean Ancestry	Latin American Individiuals with Mostly European and Native American Ancestry
**MMP-2**													
	−1306 C/T (rs243865) [38]	T	0.20139	0.0597	0.0608	0.025	0.073	0.055	0.129	0.12	0.24479	0.204	0.2152
	−1575 G/A (rs243866) [38]	A	0.229833	0.0562	0.0574	0.022	0.087	0.079	0.1594	0.112	0.242161	0.203	0.2182
**MMP-9**													
	−1562 C/T (rs3918242) [38]	T	0.17402	0.2275	0.2330	0.070	0.226	0.267	0.327	0.04	0.16158	0.110	0.084
	+279 A/G (rs17576) [38]	G	0.355297	0.34381	0.34448	0.324	0.755	0.755	0.4824	0.755	0.355310	0.3325	0.2256

**Table 2 ijms-23-10571-t002:** Comparison of the frequency of genotypes and alleles of individual polymorphisms of MMP-2 and MMP-9 polymorphisms for control groups and patients with type 2 diabetes.

Gelatinase	SNP	Control, n (%)	Patients, n (%)	χ2	*p*-Value	OR (95%CI)
**MMP-2**						
S. Sarray et al. (2021) [29]	−1306 C/T (rs243865)	Σn = 310	Σn = 791			
	genotype					
	CC	(64.1)	(73)		0.006	1.00
	CT	(27.3)	(23.3)			0.75 (0.54–1.03)
	TT	(8.6)	(3.7)			0.38 (0.21–0.7)
	CT + TT	(35.9)	(27)			0.66 (0.49–0.89)
	CC + CT	(91.5)	(96.3)		0.003	1.00
	allele frequency					
	C	(77.75) *	(84.65) *			
	T	(22.25) *	(15.35) *			

S. Sarray et al. (2021) [29]	−1575 G/A (rs243866)	Σn = 310	Σn = 791			
	genotype					
	GG	(66.1)	(74.1)		0.012	1.00
	GA	(30.4)	(23.7)			0.70 (0.51–0.95)
	AA	(3.5)	(2.2)			0.56 (0.25–1.25)
	GA + AA	(33.9)	(25.9)			0.68 (0.51–0.92)
	GG + GA	(96.5)	(97.8)		0.25	1.00
	allele frequency					
	G	(81.3) *	(85.95) *			
	A	(18.7) *	(14.05) *			

**MMP-9**						
K. Singh et al. (2013) [39]	−1562 C/T (rs3918242)	Σn = 267	Σn = 353			
	genotype					
	CC	196 (73.40)	204 (57.8)			Ref.
	CT	69 (25.84)	137 (57.8)	13.28 *	0.00027 *	1.91 (1.34–2.71) *
	TT	2 (0.75)	12 (3.4)	6.53 *	0.01061 *	5.76 (1.27–26.09) *
	CT + TT	71 (26.6)	149 (42.2)	16.2 *	0.00006 *	2.02 (1.43–2.84) *
	allele frequency					
	C	461 (86.33)	545 (77.2)			Ref.
	T	73 (13.67)	161 (22.8)	16.57 *	0.00006 *	1.87 (1.38–2.53) *

S. Saravani et al. (2017) [40]	+279 A/G (rs17576)	Σn = 120	Σn = 120			
	genotype					
	GG	102 (85)	84 (70)			Ref.
	AG	18 (15)	36 (30)	7.74 *	0.00540*	2.43 (1.29–4.58) *
	AA	0	0		1.00000	
	allele frequency					
	G	222 (92.5)	204 (85)			Ref.
	A	18 (7.5)	36(15)	6.76 *	0.00932 *	2.18 (1.2–3.95) *

* Values calculated based on the data presented by the authors.

**Table 3 ijms-23-10571-t003:** Characteristics of control groups and patients with type 2 diabetes.

			Control Subjects			T2D Patients			
Study		Ethnicity	Gender (Male:Female)	Age (Years), Mean ± SD	BMI (kg/m^2^), Mean ± SD	Gender (Male:Female)	Age (Years), Mean ± SD	BMI (kg/m^2^), Mean ± SD	Diabetes Duration (Years), Mean ± SD
**MMP-2**	S. Sarray et al. (2021) [29]	Tunisian Arab	141:169	60 ± 12.6	24.8 ± 2.8	440:351	60.6 ± 7.5	28.5 ± 5	13.6 ± 7.8
**MMP-9**	K. Singh et al. (2013) [39]	Homogeneous ethnic group of north Indian population	N/A	N/A	N/A	215:138	51.41 ± 10.56	23.89 ± 4.52	6.14 ± 5.42
	S. Saravani et al. (2017) [40]	Iranian	32:88	56.11 ± 11.075	N/A	29:91	56.57 ± 10.602	N/A	9.91 ± 6.90

**Table 4 ijms-23-10571-t004:** Gelatinase polymorphisms in diseases accompanying type 2 diabetes.

Disease	Gelatinase	SNP	Association	
**Retinopathy**				
	MMP-2	−790 T/G (rs243864)	increased risk of the disease	S. Sarray et al. (2022) [56]

		−1575 G/A (rs243866)	increased risk of the disease	S. Sarray et al. (2022) [56]

		−1306 C/T (rs243865)	doubled risk of the disease	J. Yang et al. (2010) [34]

			marginally significant increased risk of disease in males	M. Beránek et al. (2008) [57]

	MMP-9	−1562 C/T (rs3918242)	possible risk factor for the disease	K. Singh et al. (2017) [55]

**Nephropathy**				
	MMP-2	−1306 C/T (rs243865)	the presence of the C allele was associated with disease susceptibility and progression	S.R. Gantala et al. (2018) [60]

	MMP-9	+279 A/G (rs17576)	the presence of the GG genotype was independently associated with disease	C. Albert et al. (2019) [58]

		−1562 C/T (rs3918242)	the T allele was a protective factor, while the C allele contributed to the disease	S. Feng et al. (2016) [59]

			the T allele reduces the risk of disease	Z. Zhang et al. (2015) [61]
**Diabetic foot ulcers**				
	MMP-9	−1562 C/T (rs3918242)	presence of allele T leads to a higher risk of developing the disease	K. Singh et al. (2013) [39]

**Macroangiopathy**				
	MMP-9	−1562 C/T (rs3918242)	the presence of the T allele was higher in patients with disease	Y. Wang et al. (2010) [62]

**Cardiovascular comorbidity**				
	MMP-2	−1306 C/T (rs243865)	possession of the T allele was associated with a reduced risk of disease	M. Buraczynska et al. (2015) [63]


## Data Availability

The data presented in this study are available upon request from the corresponding author.

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
