# Peer review of "Association of MMP-2 and MMP-9 Polymorphisms with Diabetes and Pathogenesis of Diabetic Complications"

_ijms, 2022, doi:10.3390/ijms231810571_

Round 1

Reviewer 1 Report

Overall, this is a very well constructed review paper. It is well written, the figures look great, the tables look great, and the authors were very knowledgeable about the topics.

The only main thing to change would be the  need to re-arrange figure titles. It starts with Figure 1 (line 50), then goes to Figure 3 in the text (line 58). Please relabel this as Figure 2 in the text, figure description, and change the figure position. If this is changed, then make Figure 2 be the new Figure 3 throughout the manuscript.  

Lastly, I think the authors should write a little more about their conclusion. What do they hope to answer? What other questions can they point out in the previous literature where their research could answer some of these questions? The authors need to address some of these questions (perhaps in a small text box or a table of some kind) in order to really bring home that examining MMP-2 and MMP-9 are good targets to explore for T2D.

Reviewer 2 Report

This review article "Association of MMP-2 and MMP-9 polymorphisms with diabetes and pathogenesis of diabetic complications" by Gajwska et al. attempts to provide a perspective on the effects of genetic variants in metalloproteinases (MMP2 and 9) on T2D and its complications. However, the related association study is scarce (only one report on MMP2 and two studies on MMP9), thus this review paper did not make any sense.  

1.     Tables 1 and Table 2 are almost copied from the original papers (Refs. 34, 44 and 45).  

2.     Some sentences are described repeatedly. Sections on MMP2 (1.3) and MMP9 (1.4) could be combined and simplified.

Reviewer 3 Report

The concept and methodology of the study are appropriate. It is worth mentioning that the authors have referred to the ethnicity of the individuals when citing genotype/allele frequencies which is known to be a significant in SNP studies.

Minor issues:

1. Page 5, lines 141 – 146 – the frequency data should be in a form of a table, if not – some comment should be added to the paragraph. The same is for lines 177 – 182 – the frequency data is just stated, without additional information, e.g. Do the differences in the frequency are associated with T2D in those populations? Does the higher frequency of rare alleles of tested SNPs influence susceptibility to T2D or other traits? Is there any data on this topic? How does it refer to the study of the authors? If at all. Discussion/commentary is needed.

2. Reference [69] is cited in the text (line 210 and in Table 3) but there is no such number in the References section.

Round 2

Reviewer 2 Report

My concerns were addessed well.